# Stress Responses to Hydrogen Peroxide and Hydric Stress-Related Acoustic Emissions (MHAF) in *Capsicum annuum* L. Applied in a Single or Combined Manner

**DOI:** 10.3390/plants14162591

**Published:** 2025-08-20

**Authors:** Pablo L. Godínez-Mendoza, Amanda K. Rico-Chávez, Ireri A. Carbajal-Valenzuela, Luis M. Contreras-Medina, Rosalía V. Ocampo-Velázquez, Enrique Rico-García, Irineo Torres-Pacheco, Ramón G. Guevara-González

**Affiliations:** 1Center of Applied Research in Biosystems (CARB-CIAB), School of Engineering, Autonomous University of Querétaro-Campus Amazcala, Carr. Amazcala-Chichimequillas Km 1.0, El Marqués 76265, Querétaro, Mexico; sharkes5@hotmail.com (P.L.G.-M.); carbajalireri@gmail.com (I.A.C.-V.); miguel.contreras@uaq.mx (L.M.C.-M.); rosalia.ocampo@uaq.mx (R.V.O.-V.); ricog@uaq.mx (E.R.-G.); irineo.torres@uaq.mx (I.T.-P.); 2Facultad de Química, Universidad Autónoma de Querétaro, Campus Centro Universitario, Cerro de las Campanas, s/n, Queretaro 76010, Querétaro, Mexico; amanda.rico@uaq.mx

**Keywords:** acoustic emissions, *Capsicum annuum*, hydrogen peroxide, MAPkinases, stress

## Abstract

Hydrogen peroxide (H_2_O_2_) application in several plant species has been widely studied as a plant biostimulant; however, the use of acoustic emissions related to hydric stress (MHAF) in biostimulating plants has not been widely studied, including the response of plants to the interaction of different stress factors. The aim of the present work was to evaluate the stress response in some morphological, biochemical, and molecular variables of the single or combined application of H_2_O_2_ and MHAF in *C. annuum* L. plants. Acoustic emission frequencies were obtained in a previous study where the frequencies came from *C. annuum* plants submitted to medium hydric stress (MHAF). Our results showed that the combination of the two stressors evaluated has a possible synergistic effect on variables such as SOD activity and relative gene expressions of *ros1*, *met1*, and MAPkinases (*mkk5*, *mpk4-1*, *mpk6-2*), as well as an antagonistic effect for flavonoid content, DPPH, and ABTS free radical inhibition, and *def1* gene expression. MHAF showed increased plant height, PAL activity, and *mpk6-1* and *erf1* gene upregulation, while H_2_O_2_ increased POD activity and upregulated *pr1a* gene. These findings suggest possible stress response pathways that are activated and enhanced by the presence of these stress factors, both individually and in conjunction with one another, making it possible to use them as novel strategies for agricultural stress management.

## 1. Introduction

Pepper (*Capsicum* spp.) is a plant species and popular condiment around the world. It is part of the *Solanaceae* family and has economic importance in different countries, such as Mexico, where *Capsicum annuum* L. is the main cultivated species [1]. Its applications in cuisine are widely known; however, it also has different applications due to compounds with anticancer, anti-inflammatory, antimicrobial, and antioxidant properties. Between these compounds, there are carotenoids, flavonoids, essential oils, and capsaicinoids [2]. While many other vegetables may contain the same compounds, capsaicinoids are a group of alkaloids unique to the *Capsicum* genus, which provide the pungency of the pepper [3]. While there are several capsaicinoids, such as capsaicin, dihydrocapsaicin, nordihydrocapsaicin, homocapsaicin, and homodihydrocapsaicin, the most prevalent is capsaicin with a relative abundance of 69% in comparison to other capsaicinoids [4]. In the context of agricultural strategies for growth promotion, some capsaicinoids involve the use of toxic agents that result in adverse effects on environmental and human health integrity [5]. At the same time, some places where *Capsicum* is grown might present drought risk, which greatly damages these crops [6]. In this context, there is interest in using controlled elicitation strategies, including different stress factors, in search of biostimulant effects that have no environmental risk as an alternative to traditional methods [7] and studying agricultural practices in order to manage biotic and abiotic stress tolerance in *Capsicum* crops [8].

Different stressors have been studied for their implementation in agricultural practices to promote growth and enhance biotic and abiotic tolerance of crops [8]; however, although these stressors have displayed positive effects for plant growth and development in pepper, there is a lack of research regarding the combined interaction of stressors. One such stress factor is hydrogen peroxide (H_2_O_2_), which is an important reactive oxygen species (ROS) that functions as a signaling molecule for oxidative stress, activating signaling cascades as a result [9]. On the one hand, its foliar application can result in the attenuation of symptoms of the pepper golden mosaic virus [10]. In another study, H_2_O_2_ foliar application in doses between 0 and 400 mM in combination with low fertigation of *C. annuum* in a hormetic scheme resulted in the biostimulation of crop development and growth [11]. On the other hand, a novel strategy that is being studied is the use of acoustic emissions in specific vibration frequencies as stressors in plants. These acoustic frequencies are defined as an oscillation of pressure waves that can be transmitted through a medium such as a liquid, solid, or gas; these waves transport energy through their propagation [12]. They are referred to as stress factors since they trigger transduction cascades, as other abiotic stress factors [13]. For example, the acoustic frequencies that a caterpillar produces while chewing have been applied to plants, resulting in a higher production of anthocyanin as well as endogenous regulation of phytohormones, similar to if the plant were in the presence of a pathogen [14]. Moreover, a study conducted with *C. annuum* plants identified specific frequencies emitted by these plants when they were submitted to low, medium, and high hydric stress levels (water deficit) [15]. These latter frequencies were applied to well-watered *C. annuum* plants, and interestingly, application of the frequencies related to medium hydric stress treatment (MHAF) was the most effective at inducing plant immunity features [16]. Both stress factors were studied individually by our research group.

It is well known that the development and response of plants to environmental cues are tightly regulated by signaling networks. Biochemical responses are triggered in the presence of stress factors, resulting in changes in the expression of enzymes such as phenylalanine ammonia lyase (PAL), superoxide dismutase (SOD), catalase (CAT), and peroxidases (POD), reactive oxygen species (ROS) accumulation, and antioxidant capacity [17]. A highly important signaling network is the mitogen-activated protein kinase (MAPK) cascade, which comprises the following in hierarchical order: MAPK kinase (MAPKKK), MAPK (MPKK), and MAPK [18]. While there are several characterized MAPKs in different plant species, there has been a lack of study on peppers. Research has characterized two MAPKs that can be induced by becoming wounded, UV-C, and cold treatment [18]. In addition, another group was able to identify a total of 19 MAPK genes and five MAPKKs; they observed different transcriptional expressions in different organs and in the responses of pepper to different biotic and abiotic stresses, meaning that MAPKKs might have importance in the regulation of pepper growth, development, and responses to stress [19]. The expression analysis of MAPK families in pepper is of interest to conduct further functional investigations on these signals. In the case of defense-related genes, several of them are regulated when the plant is in the presence of a stress factor. These include pathogenesis-related (*pr*) genes, which are involved in responding to both biotic and abiotic stresses [20]. Stressors induce the expression of genes involved in the establishment of acquired systemic resistance (SAR) and induced systemic resistance (ISR), the first of which was established by genes, such as *npr1* and *pr1*, while the second was established by transcription factors such as *erf* [21]. Other genes such as repressor of silencing 1 (*ros1*), that result in DNA demethylation, play an important role in regulating defense pathways in response to stressors [22].

Currently, there is little information about the effects of MHAF as an abiotic stressor, and there is no information regarding its effect on plants over the signaling network they interact with, such as MAPK and defense responses. Additionally, to our knowledge, there are no studies involving the combination of MHAF with other stress factors (such as H_2_O_2_). This aspect is important because there is an increasing interest in evaluating the interaction between two or more stressors in plants since there are few studies on this; although the likelihood of both stressors affecting the plant in normal cultivation is unlikely because of the specific frequencies used, their application is easy and it is feasible to test them in agricultural practice. Based on the abovementioned findings, the aim of the present work was to evaluate the stress response in morphological, biochemical, and molecular variables of the single or combined application of H_2_O_2_ and MHAF in *C. annuum* L. plants. Our results provide basic information to further understand the effect of these stress factors on applications, such as agricultural sustainable biostimulant treatments.

## 2. Results

With regards to *C. annuum* plant height, Figure 1 shows that there were statistical differences among treatments in comparison to the control. On the one hand, H_2_O_2_ treatment was not statistically different from the other groups; however, both MHAF and MHAF+H_2_O_2_ treatments were significantly higher than the control group. On the other hand, the stem width of treatments did not differ from each other.

In Figure 2, a general representation of the aerial and root parts of the plants treated with H_2_O_2_ and MHAF in single or combined applications is shown.

### 2.1. Biochemical Responses

#### 2.1.1. Capsaicin and Endogenous Hydrogen Peroxide

The total capsaicin content obtained with gas chromatography–mass spectrometry (GC-MS) showed a difference between the control (with a higher concentration) and the other treatments, including the H_2_O_2_, MHAF, and combination treatments (MHAF+H_2_O_2_) (Figure 3). In the case of the endogenous H_2_O_2_ content, a significant difference was observed for every treatment compared to the control group (Figure 3). This latter result was expected since the treatments are stress factors, which are known to increase the production of endogenous H_2_O_2_.

#### 2.1.2. Antioxidant Enzymatic Activity

The results obtained from enzymatic assays are shown in Figure 4. The assay for PAL indicated that there was a higher activity when the plants were treated with MHAF and the combination MHAF+H_2_O_2_ by almost three-fold. In the case of SOD assays, the results indicate that only the combined treatment presented significantly higher activity, and interestingly, the H_2_O_2_ and MHAF treatments applied individually resulted in significantly lower activity compared to the control group. In the case of CAT activity, all the treatments displayed significantly lower activity compared to the control group. POD activity displayed the fact that H_2_O_2_ and the combined treatments showed significantly higher activity than the control, with the highest result obtained in the H_2_O_2_ treatment. These results suggested that under the conditions of our research, POD activity might have a more significant antioxidant role than CAT in the application of these stressors when attempting to scavenge ROS produced by the treatments in the chili plants studied.

#### 2.1.3. Total Phenolics, Total Flavonoids, and Antioxidant Capacity

In Figure 5, the content of total phenolics and flavonoids, as well as the antioxidant capacity in the leaves of the studied plants, is shown. On the one hand, for total phenolic content, there was no statistically significant difference between the treatments; however, for the total content of flavonoids, the MHAF treatment was significantly (three-fold) higher than the MHAF+H_2_O_2_ treatment. On the other hand, for the antioxidant capacity, DPPH and ABTS assays were conducted. In the case of the DPPH assay, only the combined treatment was different than the other treatments, presenting a lower inhibition percentage and, thus, lesser antioxidant capacity with this assay. For the ABTS assay, the combined treatment had the same inhibition percentage as the control group; however, individual treatments of H_2_O_2_ and MHAF had higher antioxidant capacity than the other groups.

### 2.2. Molecular Responses

#### 2.2.1. Expression of MAPkinases

In Figure 6, the expression of different MAPkinases genes can be observed with the previously mentioned treatments. The genes analyzed were *mkk5*, *mpk4-1*, *mpk4-3*, *mpk6-1*, and *mpk6-2*. Of these, *mkk5* is an MAPkinase, while the rest are MAPkinases. For *mkk5*, the combined treatment induced higher expression in comparison to the control and H_2_O_2_ treatment; this response was the same for the *mpk6-1* gene. In the case of *mpk4-1* and *mpk6-2*, the combined treatment was statistically significantly higher than every other group. Alternatively, for the *mpk4-3* gene, the control group had a higher expression, while the MHAF and combined treatment presented a lower expression.

#### 2.2.2. Expression of Induced Systemic Resistance (ISR) Gene Markers

The expression of genes related to defense, *erf1*, *asr1*, and *def1* was evaluated via ISR. The results showed that there were statistical differences regarding the *erf1* gene in MHAF treatment, which displayed the highest expression. Similarly, for the *asr1* gene expression, MHAF had the highest expression; however, H_2_O_2_ was also upregulated compared to the control, while the combined treatment displayed downregulation. In the case of the *def1* gene, there was a significant difference when combining the treatments, with H_2_O_2_ treatment alone between the combination and the control. The results are shown in Figure 7.

#### 2.2.3. Expression of Systemic Acquired Resistance (SAR) Gene Markers and Genes Related to DNA Methylation/Demethylation

The gene expression of some SAR gene markers was evaluated (Figure 8). Expression of genes related to DNA methylation/demethylation (ros1 and met1) was analyzed. For both genes, MHAF+H_2_O_2_ treatment resulted in higher expression in comparison to the other groups. The results showed that npr1 was upregulated by MHAF as well as the combination of treatments, with the latter showing the highest expression. However, for the pr1a gene, H_2_O_2_ was significantly higher than the control and MHAF, while the combined treatment resulted in a higher expression.

### 2.3. Multivariable Analysis

A multivariable analysis of principal components (PCA) was conducted in order to visualize correlations between treatments and the different evaluated variables in the study in more detail (Figure 9); in addition, an ANOVA test was performed to determine the effect of interactions of the two stressors evaluated (Appendix A) and was made to confirm possible interactions between treatments. The results showed a trend of several genes towards the combined treatments.

## 3. Discussion

### 3.1. Morphological Response

In the present study, in accordance with the study of Caicedo-Lopez, where they used chili pepper (type Jalapeño), a higher plant height was obtained in the present study for the pepper type Serrano. While there was no difference with the H_2_O_2_ treatment, the combination of stress factors, as well as the individual MHAFs, resulted in a significant increase in height in comparison to the control. These results suggest that MHAF induces growth and, similar to those frequencies originally obtained from a Jalapeño-type pepper, a corresponding effect was also observed in Serrano pepper.

### 3.2. Capsaicin and Hydrogen Peroxide

A significantly higher concentration of endogenous H_2_O_2_ was observed in every treatment in comparison to the control. This result was expected since the application of stress in plants generates ROS; in this context, H_2_O_2_, despite being a ROS, depending on the concentration, was also an important signaling molecule playing a vital role in plant growth and development, as well as the coordination of responses against stress [23]. For the total capsaicin content, there was a higher concentration in the control compared to the treatments; however, the research of Caicedo-Lopez [16] reported higher capsaicin production. The difference in results between that study and the present work might be due to the different chili pepper types studied; moreover, another explanation might be that the higher activity of POD observed in the present work diminished the content of oxidized capsaicin in the analyzed samples, as suggested by other authors [24].

### 3.3. Antioxidant Enzymatic Assays

Notably, SOD, CAT, and POD are important antioxidant protective enzymes in plants, since they play a vital role in the clearance of reactive oxygen species (ROS) from plant systems [25]. In the case of enzymatic assays, an increase for both MHAF and the combined treatments was observed for PAL activity. The MHAF result is in accordance with the results of Caicedo-Lopez [16], while the H_2_O_2_ treatment showed no increase compared to the control group; this is in accordance with a study reported by Mejía-Teniente [9]. PAL is a key enzyme in phenylpropanoid metabolism that is usually associated with plant disease resistance [26], suggesting that MHAF induces plant defense, at least in part, by increasing the activity of PAL. In the case of SOD, both individual stress factors presented lower activity than the control; however, the combination of treatments showed a higher enzymatic activity than the control group. This latter result suggests that the interaction of both stress factors has a synergistic effect on SOD activity in the type of *C. annuum* evaluated. While in the present study, less CAT activity was observed with all the treatments compared to the control group, an increase in POD activity was also detected. Taken together, these latter results suggest that, while CAT activity was low, the antioxidant enzymatic protective mechanism was likely (at least partially) compensated in the plant system by increasing POD activity, and perhaps by other antioxidants not measured in the study (i.e., glutathion), which could be responsible for fully compensating the antioxidant system. It is noteworthy that H_2_O_2_ treatment displayed the highest activity of POD, and the combined treatment resulted in values between both individual stress factors, suggesting that MHAF acted antagonistically to H_2_O_2_ treatment for this biochemical variable.

### 3.4. Antioxidant Non-Enzymatic Assays

On the other hand, although the phenolic content did not show any differences, the content of the flavonoid showed a significantly increasing level when applied to MHAF treatment, only in contrast to a combination of stressors. Both H_2_O_2_ and MHAF treatments were higher than the combination of stressors, suggesting that there was an antagonistic effect between H_2_O_2_ and MHAF in flavonoid production. Similar responses were also observed in the antioxidant capacity assays of DPPH and ABTS, where the combined treatment displayed a lower inhibition percentage. The results in both assays differed in the control group. While there was no difference among the individual treatments in the DPPH assay, in the ABTS assay, the control group had a significantly lower value than the individual treatments, resulting in a lack of correlation between assays due to the difference in the nature of the reaction mechanisms of both assays [27,28]. These results suggest that immunity markers expressed in the presence of these stress factors are likely more related to enzymatic antioxidant activity at a biochemical level, trying to scavenge ROS levels generated by the treatments. Based on antioxidant indicators, combined stress may lead to more severe oxidative damage.

### 3.5. Relative Gene Expression

To our knowledge, this is the first study that has been carried out on the response of MAPkinases with acoustic emissions as stressors, shedding light on the signaling cascade route that responds to the evaluated acoustic emissions. The results showed that the genes *mkk5*, *mpk4-1*, *mpk6-1*, and *mpk6-2* were upregulated with the combined treatment, while *mpk4-3* was downregulated by individual stressors as well as the combination treatment, suggesting a synergistic effect between both stressors for the expression of the four upregulated genes. This result is in accordance with Lui [19], where expression of *mkk5* was related to the other MAPkinases mentioned above. In that research, several stress treatments were applied; however, the use of H_2_O_2_ and MHAF was not tested. Our results showed that the evaluated stressors, when combined, could activate a pathway with *mkk5*, following the other tested MAPkinases apart from *mpk4-3*, suggesting that *mpk4-3* might be triggered by other stress factors (or perhaps another AF, dB, or even H_2_O_2_ dose and combination). While this may have different effects on different species, the decibel (dB) and frequency results obtained in the present study strongly suggest that acoustic emissions in combination with H_2_O_2_ might trigger a pathway through *mkk5* before proceeding through the other three MAPkinases evaluated in the present work.

*erf1* and *asr1* are genes related to the induced systemic resistance (ISR) defense pathway, which helps plants to tolerate biotic and abiotic stresses, usually involving ethylene and jasmonic acid signaling pathways, respectively [21,29]. Our results showed that the *erf1* gene was upregulated by MHAF; however, for the *asr1* gene, the combination of stressors and H_2_O_2_ upregulated its expression. Interestingly, the other ISR gene marker evaluated (*def1*), which is involved in DNA damage resistance and genomic maintenance [30], displayed an unregulated expression with the individual treatment application, while the combined treatment was downregulated compared to the control group, suggesting that the stressors evaluated had an antagonistic effect for this molecular variable.

DNA demethylation can be performed using the *ros1* gene [31]. At the same time, *met1* is a gene encoding a DNA methyltransferase and is in charge of performing de novo methylation in order to maintain the methylation of DNA [32]. These genes help to regulate DNA expression by silencing or activating the expression of different genes depending on environmental stimuli. Abiotic stress can affect dynamic changes in DNA methylation; these changes regulate the expression of genes related to stress response, improving plant tolerance to stress [33]. In this context, we observed that the combined stress factors upregulated both *ros1* and *met1* genes, suggesting a synergistic effect when combining stress factors. These genes play an important role in defense pathways in response to abiotic stress [22]. Notably, *ros1*, contrary to other genes, was upregulated when methylated, which suggests that while *met1* silences some genes, the upregulation of *ros1* by *met1* could activate or upregulate other genes when applying one or both of the stress factors evaluated in this study, as was seen with the other genes analyzed in this research. Interestingly, in another study [34] carried out with tobacco plants (*Nicotiana tabacum* cv. Xanthi) treated with foliar applications of H_2_O_2_, progeny hypomethylation was shown in genes involved in the cellular response to environmental stimuli, suggesting the induction of intergenerational memory. Thus, studying the gene expression of DNA methylases and demethylases, in addition to other epigenetic components in plants, could help unravel cues in stress memory in plants.

The genes *npr1* and *pr1a* are related to plant defense, specifically the systemic acquired resistance (SAR) defense pathway, which is this plant’s wide-spectrum immune response against pathogens [20,35]. For the *npr1* gene, results showed that MHAF was the stressor responsible for its upregulation, while results for *pr1a* showed that there was a significant difference in expression when using H_2_O_2_ individually; however, MHAF was downregulated compared to the control. While the combination treatment produced a higher expression for both genes, there was a modest difference between their respective individual responsible stressors and the combined treatment, suggesting that for these variables, each of them has a different responsible stressor for upregulation. This also suggests that *C. annuum* plants treated with combined stressors might be more tolerant to pathogen attacks, which is a theory that should be tested in future studies.

### 3.6. PCA Interpretation

For the PCA results carried out (Figure 9), the data suggested that the combined treatment displayed a greater correlation with multiple variables. This combined treatment showed correlation with variables for the relative expression of genes such as MAPkinases, except for the *mpk4-3* gene, as well as genes such as *npr1*, *ros1*, and *met1*, which is in accordance with the individual analysis of these genes, which suggested a synergistic effect. In the same way, a correlation was observed between PAL activity and MHAF treatment. For *mpk4-3*, the data indicated that treatments had a negative regulation compared to the control group, suggesting that these stress factors did not activate the pathway that *mpk4-3* followed; however, this was confirmed for the rest of the MAPkinases analyzed. Also, data for H_2_O_2_ treatment suggested that there was relevance towards POD and *pr1a*. On the other hand, MHAF was relevant for plant height, PAL, *mpk6-1*, and *erf1*. However, for the combination of stress factors, the results showed an antagonistic interaction for flavonoids, DPPH, ABTS, and *def1*, while there was a synergistic effect for SOD, *mkk5*, *mpk4-1*, *mpk6-2*, *ros1*, and *met1*.

## 4. Materials and Methods

### 4.1. Chemical Reagents

Solvents of analytical HPLC grade were obtained from Baker (Mallinckrodt Baker, Inc., Phillipsburg, NJ, USA). Gallic acid, quercetin, salts, Folin–Ciocalteu, DPPH, and ABTS reagents were obtained from Sigma Chemical Co. (St. Louis, MO, USA).

### 4.2. Capsicum annuum *L.* Cultivation

Seeds (Starseeds, Puebla, México) of serrano chili (*C. annuum*) were planted and germinated in a 200-hole tray. The substrate used had a composition of 70% peatmoss and 30% vermiculite. The substrate was watered to field capacity. After the seedlings acquired their first true leaves, they were transferred to individual plastic bags with a substrate composed of 60% tezontle, 15% sand, and 25% peatmoss. The bags with the seedlings were placed in a greenhouse with four trays, each containing 20 plants. The greenhouse was equipped with an automatic irrigation system that watered the plants with 500 mL of a Steiner solution daily [36]. The greenhouse was equipped with a sound-mitigating wall in the middle of the structure. Two trays were placed on each side of the greenhouse (Appendix A). The greenhouse was located at the Autonomous University of Querétaro, Campus Amazcala, El Marqués, Querétaro, México (20°70′55.3″ N, 100°25′93.09″ Q). The environmental conditions in the greenhouse during the study were an average temperature of 18.8 °C, relative humidity of 64.6%, and light duration (70–85 μmol m^−2^ s^−1^ of photosynthetically active radiation) of 13 h. The application started when the plants were two months old.

### 4.3. MHAF Application

Two speakers (Yamaha, NS-IC 600) (120 W, frequency response of 65 Hz–28 kHz) were used perpendicular to one of the trays. The treatment was set at 69 dB, and a sonometer (EXTECH SL510) was used to accommodate the volume of the speakers to 69 dB at the tray’s position. Frequencies of *C. annuum* associated with medium hydric stress emission patterns [16] were used as the treatment (higher amplitude sound files for Mp3: 240 Hz, 320 Hz, 480 Hz, and 525 Hz; amplitude velocity range: 0.0254–1.13 mm s^−1^). The trays with the speakers were placed close to each other, taking care that they received the treatment at 69 ± 1 dB. The other two trays were placed at the other side of the greenhouse’s sound-mitigating wall and received 20 ± 1 dB less intensity; the wall was made of 18 mm plywood and covered with acoustic foam panels to mitigate the MHAF towards the other side of the greenhouse. Applications were carried out in 20 min sessions once every two weeks. Plants that would receive the individual and combined treatments were selected randomly. Environmental noise interference was mitigated by the randomly placed control groups in the trays, meaning that the environmental noise would affect equally the control and treatment groups.

### 4.4. Hydrogen Peroxide Application

A solution of 50 mM of H_2_O_2_ was prepared at the time of application as suggested by ref. [11]. Foliar application was carried out to the dripping point once every two weeks. Plants that would receive the individual and combined treatment were selected randomly. Combined applications were carried out, altering the stressor each week.

### 4.5. Morphological Variables Measurement

Once per week, morphological measurements were recorded for each group, each consisting of 20 plants. The height was obtained with the help of a metallic ruler, while the width of the stem was measured with an electronic vernier caliper. After the last application, ten randomly selected leaves and fruits were removed from each plant, frozen with liquid nitrogen, and stored at −40 °C until analysis.

### 4.6. Enzymatic Assays

The preparation of sample extracts from leaves was obtained by grinding them with a mortar and pestle with the help of liquid nitrogen, and 0.3 g of the sample was transferred to an Eppendorf tube and mixed with 1 mL of extraction buffer (KH_2_PO_4_ 0.05 M, pH 7.8) as described by Hayat [37]. A vortex was used to mix the samples with the buffer for two minutes, and this was centrifuged for 15 min at 13,685× *g* at 4 °C. Lastly, the supernatant was transferred to another tube and maintained at 4 °C until use.

A calibration curve with cinnamic acid for the PAL assay was made using an absorbance value of 290 nm. Microplates of 96 wells were used, where the reaction consisted of 230 µL of the reaction buffer (Borate 0.1 M, L-phenylalanine 10 mM, pH 8.8) and 20 µL of sample extract in triplicate. The plate was then incubated at 40 °C for 1 h. To finish the reaction, 50 µL of HCl 1N was added to each well, and the plate was left resting for 10 min. A spectrophotometer (Thermo Fisher Scientific, Mexico City, Mexico; MULTISKAN Sky High) was used to read the samples at 290 nm. The results were reported as U mg^−1^ protein [38].

Eppendorf tubes of 2 mL were used for the SOD assay; each reaction consisted of 0.75 mL of a reaction buffer (KH_2_PO_4_ 0.05 M, pH 7.8), 0.15 mL of EDTA 0.1 mM, 0.15 mL of methionine 0.13 M, 0.15 mL NBT 0.75 mM, 0.15 mL riboflavin 0.02 mM, 0.075 mL extract of sample, and 0.075 mL of distilled water. Inversion was used to mix the reaction tubes, which were then exposed to light for 30 min. By triplicate, 300 µL of the reaction was transferred to a well in a microplate, reading at an absorbance of 560 nm. The results were reported as U mg^−1^ protein [37].

For the CAT assay, three solutions were prepared. A cobalt (II) solution was prepared with 253.5 mg of CO(NO_3_)_2_ in 12.5 mL of distilled water, a Graham salt solution was prepared with 125 mg of (NaPO_3_)_6_ and 12.5 mL of distilled water, and lastly a sodium bicarbonate solution was prepared with 11,250 mg of NaHCO_3_ and 125 mL of distilled water. These three solutions were combined to prepare the working solution. First, 6.25 mL of cobalt (II) was mixed with 6.25 mL of the Graham salt solution until completely homogenized; then, 112.5 mL of sodium bicarbonate was added to the mixture. Separately, a solution of H_2_O_2_ 10 mM was prepared with a phosphate buffer (50 mM and pH 7.0). The assay was carried out by preparing a standard with 75 µL of distilled water and 150 µL of H_2_O_2_; the samples contained 75 µL of the extract with 150 µL of H_2_O_2_. Each tube was vortexed and incubated for 2 min at 37 °C. After this, 900 µL of a freshly made working solution was added to each tube, mixed by vortex, and maintained at room temperature in the dark for 10 min. By triplicate, 300 µL of each tube was transferred to a well in a microplate, and their absorbances read at 640 nm. The results were reported in U mg^−1^ protein [39].

For the POD assay, a reaction mix was made, containing 0.15 mL of 16.3 mM guaiacol, 0.15 mL of 900 mM H_2_O_2_, 2.66 mL of 0.1 M phosphate buffer pH 7.0, and 40 µL of sample extract, and the absorbance was read at 470 nm. The reaction mix was conducted in triplicate for each sample. The results were expressed as U mL^−1^ sample [40].

Protein determination was carried out using a calibration curve with bovine serum albumin, reading the absorbance at 595 nm. The assay consisted of mixing 20 µL of the extract with 230 µL of Bradford reagent in a well in a microplate in triplicate. The samples were maintained in the dark at room temperature for 20 min and their absorbances were read at 595 nm [41].

### 4.7. Capsaicin Quantification

Samples were lyophilized and ground with a mortar and pestle; then, the powder was extracted with acetonitrile (1:20; *w*/*v*) and transferred to an ultrasonic bath for 3 h at 30 °C. Afterwards, the samples were agitated at 7× *g* for 12 h and centrifuged at 12,000× *g* at 10 °C for 10 min. Once more, the acetonitrile extraction procedure was repeated, and the supernatants were collected and dried with nitrogen (N_2_) stream. The extracts were resuspended with 200 µL of acetonitrile and centrifuged at 12,000× *g* at 10 °C for 10 min for gas chromatography analysis (GC) in an Agilent 5975 chromatograph (Agilent Scientific Instruments, Mexico City, Mexico) equipped with a quadrupole mass detector (Agilent, 2975C). For the mass spectrometer, electron energy was set at 70 eV, and a dual acquisition in both the full SCAN and SIM modes was set with mass range at m/z 29–500 and a 100 ms dwell time targeting the 137 m z^−1^ ion. The separation was carried out with an HP-5MS column (30 m × 0.25 mm, 0.25 µm). For 1 min, the initial oven temperature was maintained at 90 °C; then, the temperature was raised to 210 °C and maintained for 1 min, and afterwards it was further raised to 290 °C, where it was maintained for 1 min. Lastly, it increased to 310 °C. The temperature of the injector and the transfer line was 250 °C, with an injection volume of 2 µL. The carrier gas used was helium (He) with a flow rate of 1.2 mL min^−1^ and a 1:1 split ratio. The NIST Standard Reference Database (NIST 11) was used to compare the mass spectra to identify capsinoids. Capsaicin had a retention time of 21.8 min, and it was quantified using a 7-point external standard calibration curve, with acetonitrile as the solvent and SIM-acquired data [42]. No derivatization was used, as it would have resulted in more complex chromatograms, and the resolution of capsinoids did not change significantly whether they were derivatized or not.

### 4.8. Total Phenolic, Flavonoid Content, and Antioxidant Capacity Using DPPH and ABTS

In a mortar, 150 mg of the leaf sample was powdered with the help of liquid nitrogen to prepare the extracts. The sample was transferred to an Eppendorf tube with 1.5 mL of ethanol at 80%. Samples were sonicated for 3 min and centrifuged at 9503× *g* for 15 min at 4 °C. Lastly, the supernatant was stored at 4 °C in an Eppendorf tube until further analysis.

Total phenolic content was determined using the Folin–Ciocalteu method. A calibration curve was made using gallic acid and a microplate, and their absorbances were read at 760 nm. Results were reported as mg of the gallic acid equivalents per mg of sample [43].

For the determination of flavonoids, a calibration curve was made with quercetin. For this assay, 50 µL of sample extracts was mixed with 180 µL of distilled water, and 20 µL of 2-aminoethildiphenil borate at 1%. This was made by triplicate and carried out in a microplate; the absorbance was read at 404 nm. The results were reported as quercetin equivalents per mg of sample [44].

Antioxidant capacity was measured using two different assays. The 2,2-diphenyl-1-picrylhydrazyl (DPPH) antioxidant capacity assay was carried out by sonicating 1.5 mg of DPPH in 25 mL of methanol for 10 min in the dark. The assay was made by mixing 20 µL of the sample extract with 200 µL of DPPH solution in triplicate in microplate wells; the absorbance was read at 520 nm after 30 min of reaction. The results were reported as % of radical inhibition [45].

The reaction of the 2,2-anzino-bis(3-ethylbenzothiazoline-6-sulfonic acid) (ABTS) assay consisted of mixing 230 µL of previously prepared ABTS solution with 20 µL of the sample extract in triplicate in microplate wells; the absorbance was read at 734 nm after 30 min of reaction. The results were reported as the % of radical inhibition [46].

### 4.9. Hydrogen Peroxide Quantification

Sample leaf tissue was powdered with the help of liquid nitrogen; afterwards, 150 mg of the powdered sample was mixed with 0.25 mL TCA (0.1% *w*/*v*), 0.5 mL KI 1 M, and 0.25 mL of KH_2_PO_4_ buffer at 10 mM. The samples were centrifuged at 12,000× *g* for 15 min at 4 °C. For each sample and in triplicate, 200 µL was placed in microplate wells and incubated for 20 min at room temperature in the dark. A calibration curve was formulated with standard solutions of H_2_O_2_ in 0.1% TCA. The absorbance was read at 350 nm. The results were reported as nanomoles of H_2_O_2_ per liter of the sample [47].

### 4.10. Relative Gene Expression

Gene expression was determined by taking RNA extracts from leaf samples with the Trizol protocol (Thermo Fisher). Samples of leaves were powdered using a mortar and pestle with liquid nitrogen; afterwards, 100 mg of the powdered sample was mixed with 1 mL of Trizol reagent and vortexed for 2 min, maintaining it at room temperature for 5 min. A volume of 200 µL of chloroform was added to each sample, mixing them for 15 s, and maintaining the samples at room temperature for 5 min. The samples were centrifuged at 16,061× *g* for 15 min at 4 °C; afterwards, the supernatant was transferred to a new tube and mixed with 500 µL of cold isopropanol, maintaining the samples at room temperature for 15 min for further centrifugation at 16,061× *g* for 10 min at 4 °C. The supernatant was discarded, and the sample tubes were supplemented with 1 mL of cold ethanol at 75%; they were then centrifuged at 11,921× *g* for 5 min at 4 °C twice. The supernatant was discarded, and the samples were left to dry; afterwards, the samples were resuspended with 50 µL of TE buffer and stored at −80 °C until analysis.

cDNA synthesis was achieved using the Maxima First Strand cDNA Synthesis Kit for RT-qPCR (Thermo Fisher). Previously extracted RNA samples were adjusted to the same concentrations, and 500 ng of the RNA samples was used for the reaction. The reaction was incubated for 10 min at 25 °C and then for 15 min at 50 °C. The reaction ended by heating at 85 °C for 5 min. The samples were stored at −20 °C until analysis.

For qPCR, the Maxima SYBR Green qPCR Master Mix kit (Thermo Fisher) was used. The reaction consisted of the mixture of the reagents mentioned by the manufacturer and 500 ng of cDNA previously synthesized with modifications in the qPCR protocol. The protocol consisted of an initial denaturation of 95 °C for 3 min, denaturation of 95 °C for 30 s, annealing of 58 °C for 30 s, and extension of 72 °C for 1 min, for 60 cycles. A real-time thermocycler (CFX96 Real-Time System Bio-Rad) was used. The data was transferred to a computer for further analysis. Primers (Appendix A) were synthesized by T4 Oligo (Irapuato, Mexico). *Actin* was used as the housekeeping gene.

### 4.11. Statistical Analysis

Measurements were carried out in triplicate; the results were reported as mean values. Statistical analysis of the data was made using ANOVA followed by Student’s *t*-test (α = 0.1) for parametric data. The effect of interaction analysis was also carried out (Appendix A). In all cases, Graphpad (Prism 10.5.0) software was used. Multivariable analysis was made using RStudio (RStudio 2025.05.1) software.

## 5. Conclusions

Regarding the results obtained and supported by a statistical analysis to determine the effect of the interaction between both stressors in comparison to single ones, it was concluded that the combined effect of H_2_O_2_ and MHAF displayed a possible synergistic effect on SOD activity as well as the relative expressions of *ros1*, *met1*, *mkk5*, *mpk4-1*, and *mpk6-2*. At the same time, it showed an antagonistic effect on flavonoid content, the percentage of inhibition of free radicals in both DPPH and ABTS assays and in the *def1* gene’s relative expression. For the case of individual stress factors, MHAF was relevant for plant height stimulation and PAL activity and *mpk6-1* and *erf1* gene upregulation, while for H_2_O_2_ treatment, POD activity was increased, and the *pr1a* gene was upregulated. Following the results of MAPkinases, it was concluded that the interaction of both stress factors might follow this signaling pathway displayed by the induced genes evaluated to activate defense mechanisms against stress. However, several studies need to be carried out to confirm this aspect. Studies such as Western blot, two-hybrid test assays, and others for MAPkinases activation, as well as the determination of salicylic acid and jasmonic acid contents, should be evaluated in the future. Our results displayed the potential use for combined H_2_O_2_ in conjunction with MHAF as an environmentally friendly and easy-to-apply stress factor that could be used for the priming of plants such as *C. annuum* as biostimulants to enhance their tolerance to stressors, making them a new strategy for agricultural stress management. Future studies on crop protection and the possibility of generating a stress memory effect over the progeny of these crops will also be carried out.

## Figures and Tables

**Figure 1 plants-14-02591-f001:**
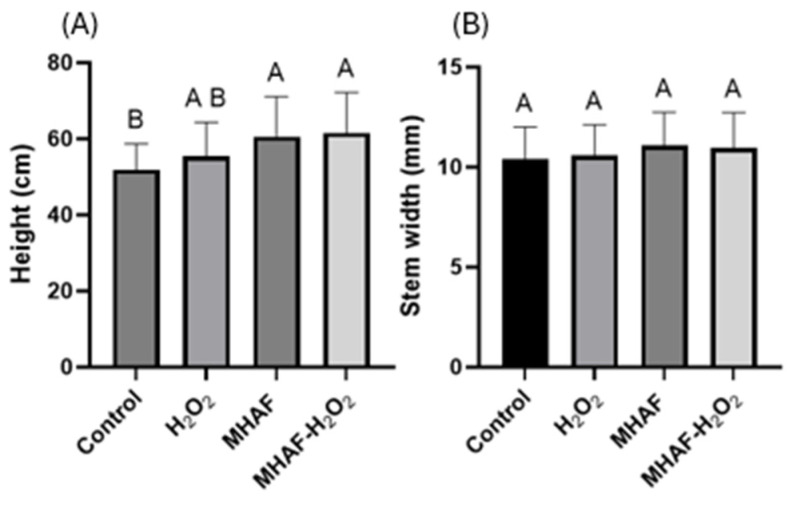
Statistical analysis of morphological variables. (**A**) Height of *C. annuum* plants treated with H_2_O_2_, MHAF, and MHAF-H_2_O_2_, expressed in cm. (**B**) Stem width of *C. annum* plants. The data is expressed as mean values. Different letters express statistical differences at α = 0.1; bars in each column indicate standard deviation; and tests were conducted in triplicate.

**Figure 2 plants-14-02591-f002:**
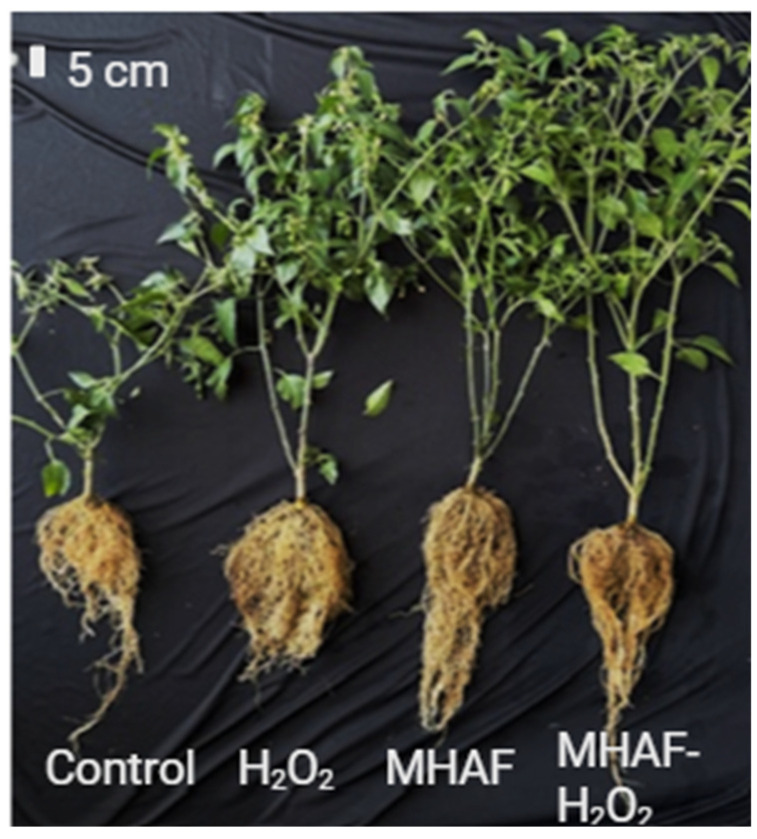
Typical phenotypic appearance of *C. annuum* L. plants in each evaluated treatment.

**Figure 3 plants-14-02591-f003:**
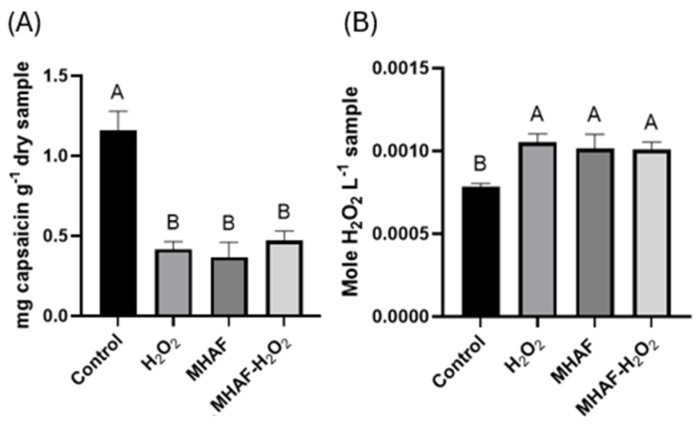
Statistical analysis of capsaicin and H_2_O_2_. (**A**) Capsaicin content from *C. annuum* fruits, treated with H_2_O_2_, MHAF, and MHAF-H_2_O_2_. (**B**) Endogenous H_2_O_2_ content from *C. annuum* leaves, treated with H_2_O_2_, MHAF, and MHAF-H_2_O_2_. Data are expressed as mean values. Different letters express statistical differences, α = 0.1; bars in each column indicate standard deviation; and tests were conducted in triplicate.

**Figure 4 plants-14-02591-f004:**
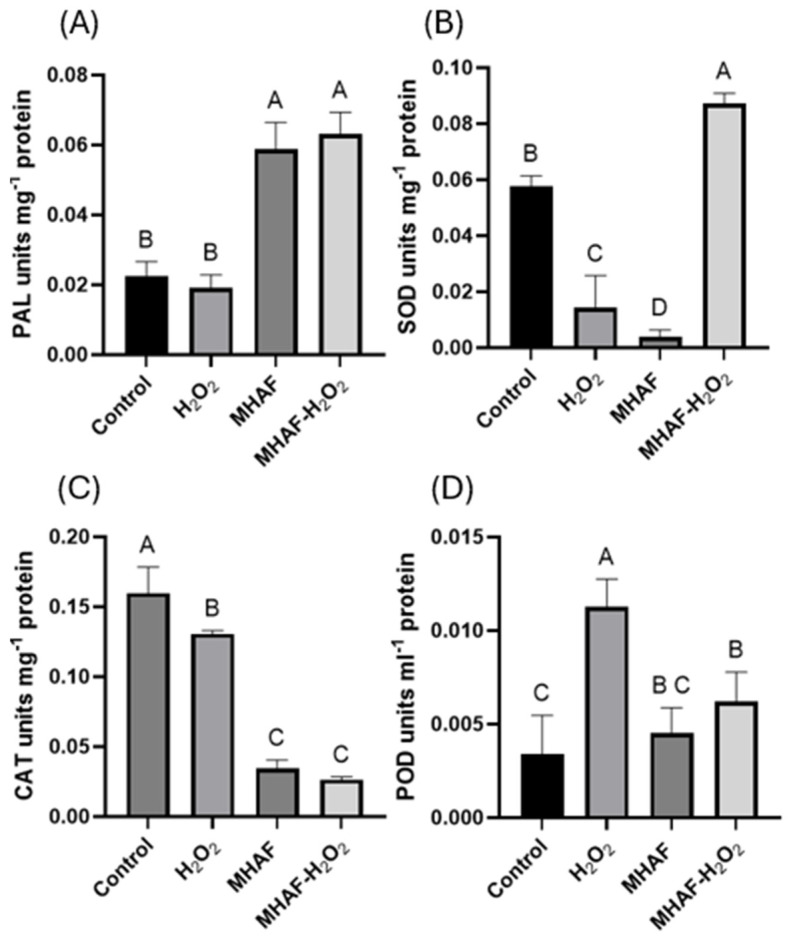
Enzymatic activity from *C. annuum* leaves treated with H_2_O_2_, MHAF, and MHAF-H_2_O_2_. (**A**) PAL activity, (**B**) SOD activity, (**C**) CAT activity, and (**D**) POD activity. Data is expressed as mean values. Within a bar, letters express statistical differences at α = 0.1; bars in each column indicate standard deviation; and tests were conducted in triplicate.

**Figure 5 plants-14-02591-f005:**
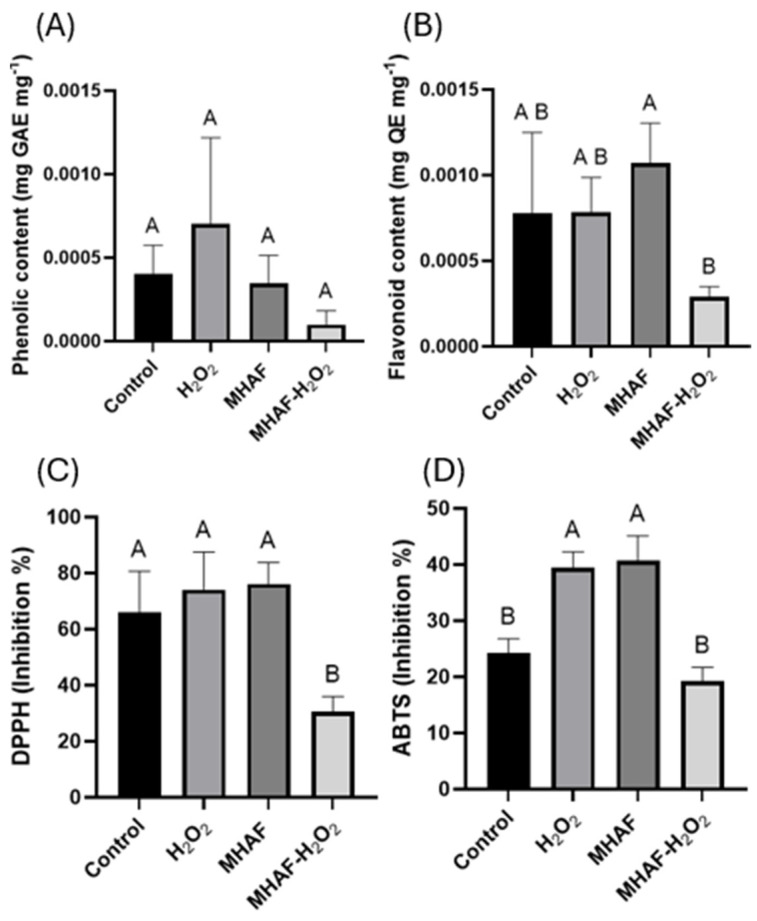
Total phenolic content, flavonoid, and antioxidant capacity with DPPH and ABTS assays from *C. annuum* leaves treated with H_2_O_2_, MHAF, and MHAF-H_2_O_2_. (**A**) Phenolic content expressed as mg of gallic acid equivalents per mg of sample. (**B**) Flavonoid content expressed as mg of quercetin equivalents per mg of sample. (**C**) The DPPH assay expressed as the inhibition percentage of free radicals. (**D**) The ABTS assay expressed as the inhibition percentage of free radicals. Data are expressed as mean values. Within a bar, letters express statistical differences at α = 0.1; bars in each column indicate standard deviation; and tests were conducted in triplicate.

**Figure 6 plants-14-02591-f006:**
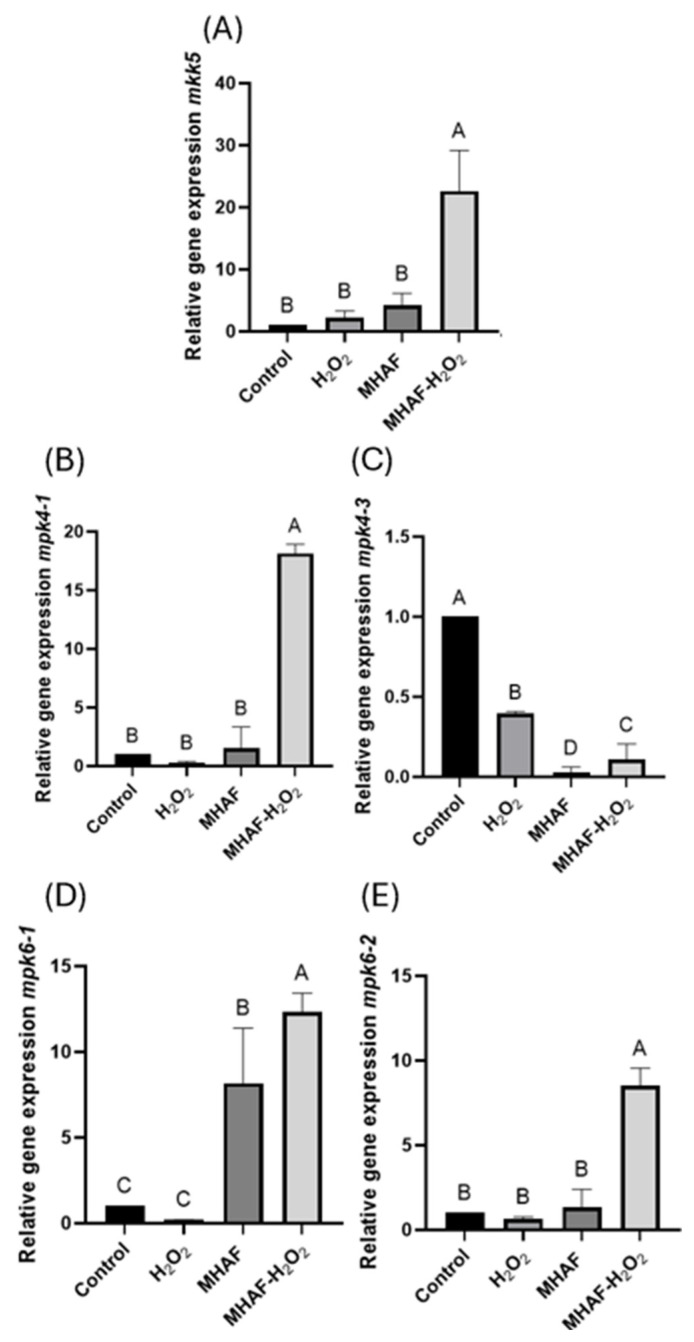
Relative gene expressions of *mkk5*, *mpk4-1*, *mpk4-3*, *mpk6-1*, and *mpk6-2* from *C. annuum* leaves treated with H_2_O_2_, MHAF, and MHAF-H_2_O_2_, using actin as a housekeeping gene. (**A**) *mkk5* gene expression. (**B**) *mpk4-1* gene expression. (**C**) *mpk4-3* gene expression. (**D**) *mpk6-1* gene expression. (**E**) *mpk6-2* gene expression. Data is expressed as mean values. Within a bar, letters express statistical differences at α = 0.1; bars in each column indicate standard deviation; and tests were conducted in triplicate.

**Figure 7 plants-14-02591-f007:**
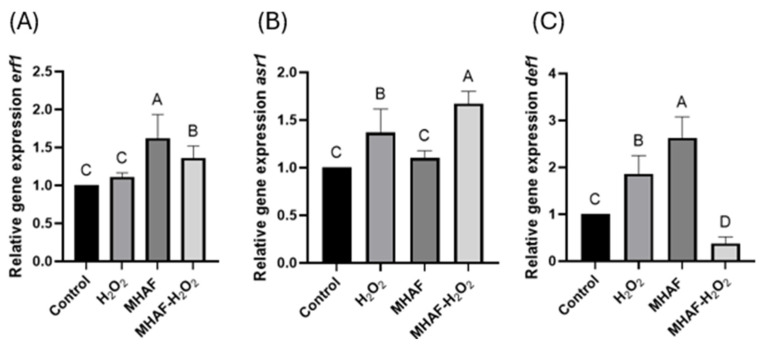
Relative gene expressions of *erf1*, *def1*, and *asr1* from *C. annuum* leaves treated with H_2_O_2_, MHAF, and MHAF-H_2_O_2_, using *actin* as a housekeeping gene. (**A**) *erf1* gene expression. (**B**) *asr1* gene expression. (**C**) *def1* gene expression. Data is expressed as mean values. Within a bar, letters express statistical differences at α = 0.1; bars in each column indicate standard deviation. Tests were conducted in triplicate.

**Figure 8 plants-14-02591-f008:**
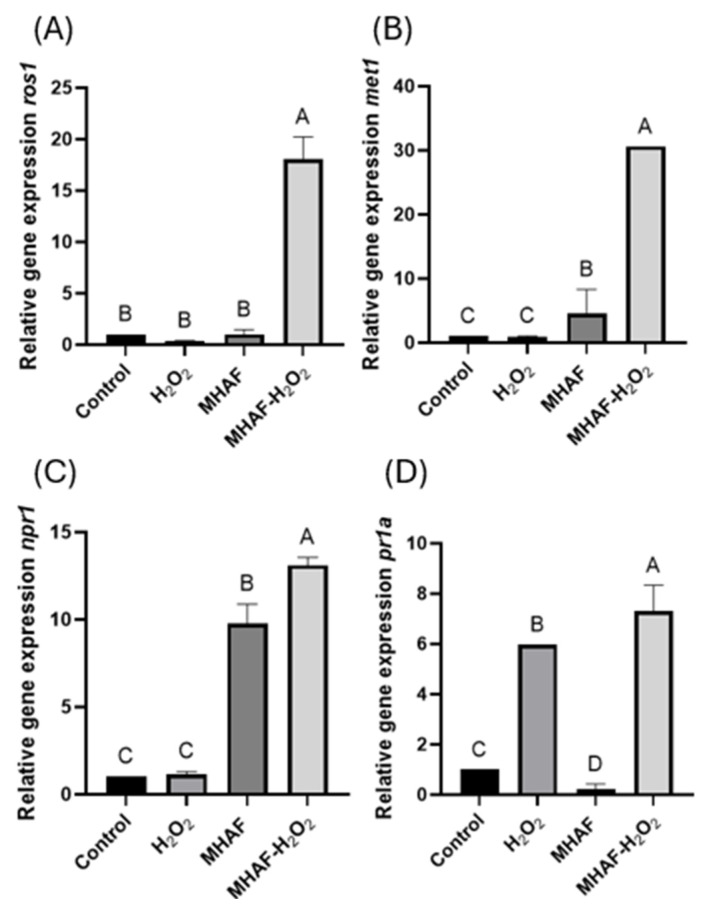
Relative gene expressions of *ros1*, *met1*, *npr1*, and *pr1a* from *C. annuum* leaves treated with H_2_O_2_, MHAF, and MHAF-H_2_O_2_, using *actin* as a housekeeping gene. (**A**) *ros1* gene expression. (**B**) *met1* gene expression. (**C**) *npr1* gene expression. (**D**) *pr1a* gene expression. Data are expressed as mean values. Within a bar, letters express statistical differences at α = 0.1; bars in each column indicate standard deviation.

**Figure 9 plants-14-02591-f009:**
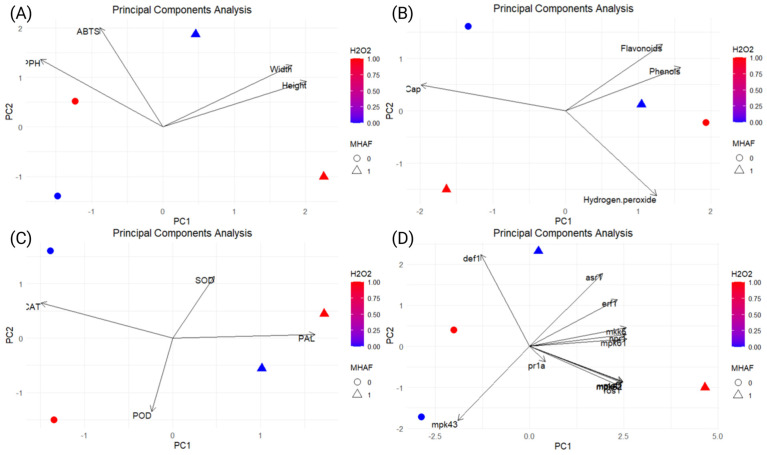
Principal component analysis (PCA) of the H_2_O_2_, MHAF, and MHAF+H_2_O_2_ treatments with the morphological, biochemical, and molecular variables evaluated in this study. (**A**) Morphological and antioxidant assays. (**B**) Phenylpropanoid compounds, capsaicin, and H_2_O_2_. (**C**) Enzymatic essays. (**D**) Relative gene expressions.

## Data Availability

The original contributions presented in this study are included in the article. Further inquiries can be directed to the corresponding author.

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
