# Peer review of "Stress Responses to Hydrogen Peroxide and Hydric Stress-Related Acoustic Emissions (MHAF) in Capsicum annuum L. Applied in a Single or Combined Manner"

_plants, 2025, doi:10.3390/plants14162591_

Round 1

Reviewer 1 Report

Comments and Suggestions for Authors

This study investigates the effects of hydrogen peroxide (H₂O₂) and moisture stress-related acoustic emission (MHAF) on the morphological, biochemical, and molecular responses of chili pepper (Capsicum annuum L.). It reveals how these stress factors, whether applied individually or in combination, influence chili growth and defense mechanisms. The findings have significant implications for developing new agricultural biostimulants that can enhance the resilience of chili peppers against stress while minimizing negative environmental impacts. However, there are several areas in this paper that require further refinement, especially in Figures 7A, B, D, and E, the errors are excessively large:

  1. The abstract lacks key results; it is recommended to include the effects of different treatments on chili pepper growth.
  2. The introduction contains excessive paragraphs and lacks a cohesive logical structure. The content of paragraphs 1 and 2 is repetitive and can be merged to streamline the background information. Similarly, paragraphs 4 and 5 are closely related and should be combined to improve the description of the current state of research.
  3. Results:

   (1) Treatment names need to be standardized; for example, the treatment names in Figures 2 and 3 should be consistent.

   (2) Figure 2 indicates that all treatments resulted in taller plants compared to the control group, but Figure 3A shows no significant differences. Was biomass measured? Can it be demonstrated that other treatments promoted chili growth?

   (3) The data reproducibility is very poor, the control error in Figure 4A is too large; and the errors in Figures 7A, B, D, and E are excessively large, retesting is recommended.

  1. In the discussion section, the beginning of paragraphs 4, 5, 6, and 7 lacks explanations of the relationships between relevant indicators and plant responses to each treatment, resulting in insufficient logic and coherence.
  2. Materials and Methods are not detailed enough:

   (1) The pepper cultivation section lacks specifications regarding light intensity and duration. Additionally, the age of the chili plants at the time of treatment application is not stated.

   (2) While the frequency and intensity of acoustic emission are described, there is no detailed explanation of how interference from other environmental noises was minimized. This treatment should logically be presented after the H₂O₂ treatment in the paper's sequence.

   (3) Section 4.5 requires specific sample quantities, such as the number of samples per treatment group.

   (4) The abbreviations DPPH and ABTS lack explanations.

Author Response

Reviewer 1:

Comments and Suggestions for Authors

This study investigates the effects of hydrogen peroxide (H₂O₂) and moisture stress-related acoustic emission (MHAF) on the morphological, biochemical, and molecular responses of chili pepper (Capsicum annuum L.). It reveals how these stress factors, whether applied individually or in combination, influence chili growth and defense mechanisms. The findings have significant implications for developing new agricultural biostimulants that can enhance the resilience of chili peppers against stress while minimizing negative environmental impacts. However, there are several areas in this paper that require further refinement, especially in Figures 7A, B, D, and E, the errors are excessively large:

The abstract lacks key results; it is recommended to include the effects of different treatments on chili pepper growth.

R- Added key results.

The introduction contains excessive paragraphs and lacks a cohesive logical structure. The content of paragraphs 1 and 2 is repetitive and can be merged to streamline the background information. Similarly, paragraphs 4 and 5 are closely related and should be combined to improve the description of the current state of research.

R- Paragraphs merged.

Results:

   (1) Treatment names need to be standardized; for example, the treatment names in Figures 2 and 3 should be consistent.

R- Names in the figures have been standardized.

   (2) Figure 2 indicates that all treatments resulted in taller plants compared to the control group, but Figure 3A shows no significant differences. Was biomass measured? Can it be demonstrated that other treatments promoted chili growth?

R- In figure 2 a general representation is shown, however, Figure 3A shows that statistical analysis displayed that MHAF and the combination had significant differences against the control. Biomass was not measured.

   (3) The data reproducibility is very poor, the control error in Figure 4A is too large; and the errors in Figures 7A, B, D, and E are excessively large, retesting is recommended.

R- The data analysed again and corrected the figures based on the retesting, error has been corrected.

In the discussion section, the beginning of paragraphs 4, 5, 6, and 7 lacks explanations of the relationships between relevant indicators and plant responses to each treatment, resulting in insufficient logic and coherence.

R- A brief explanation was added.

Materials and Methods are not detailed enough:

   (1) The pepper cultivation section lacks specifications regarding light intensity and duration. Additionally, the age of the chili plants at the time of treatment application is not stated.

R- Information added.

   (2) While the frequency and intensity of acoustic emission are described, there is no detailed explanation of how interference from other environmental noises was minimized. This treatment should logically be presented after the H2O2 treatment in the paper's sequence.

R- Since the control groups were randomly mixed with the treatment groups, environmental noise would affect the groups equally.

   (3) Section 4.5 requires specific sample quantities, such as the number of samples per treatment group.

R- quantities were added.

   (4) The abbreviations DPPH and ABTS lack explanations.

R- abbreviation explanation added.

Reviewer 2 Report

Comments and Suggestions for Authors

First of all, thank you for giving me the opportunity to evaluate this study.

The title partially reflects the content of the study.

The abstract should briefly and concisely describe the methods used in the study. Similarly, some clear data on the results should be added to the abstract.

Keywords should be listed in alphabetical order.

Line 29. The term “hot pepper” is incorrect. The scientific name for the species Capsicum annuum is pepper. The authors should have identified the plant material used in their study using one of the taxonomic classification systems accepted worldwide. For example: https://itis.gov/servlet/SingleRpt/SingleRpt

Line 67. The study mentions hydric stress-related acoustic emissions. However, it is unclear what hydric stress refers to here. Even if it has been applied in another study, the method should be clearly shared.

Line 101. In the study, hydrogen peroxide application and hydric stress-related acoustic emissions are defined as stress factors. What is the likelihood that these two different factors will affect the plant together in normal cultivation? The authors should clarify this issue.

Line 115. The photograph is not of scientific quality. Descriptions are provided from left to right. According to this, the plant on the far left is the strongest. However, no application was made to the plant on the far right, yet it is the weakest. This photograph should be reviewed.

Line 143. The statistical analysis results in Figure 4A should be reviewed. It is surprising that the control (1.5) and H2O2 (0.6) are in the same category.

The biggest shortcoming of the study is that the applications are not clearly stated. The hydrogen peroxide (50 mM) application is understandable. However, other factors are unclear. In one part of the article, there is a mention of acoustic emission associated with moderate hydric stress. In other parts, terms such as sound stress are used. This issue should be clearly explained in both the introduction and the materials and methods section. This deficiency makes it impossible to evaluate the study as a whole.

In the Conclusion section, information about the limitations of the study should be added.

 The references used in the study are appropriate in terms of both subject matter and relevance.

Author Response

Reviewer 2:

Comments and Suggestions for Authors

First of all, thank you for giving me the opportunity to evaluate this study.

The title partially reflects the content of the study.

The abstract should briefly and concisely describe the methods used in the study. Similarly, some clear data on the results should be added to the abstract.

Keywords should be listed in alphabetical order.

R- Keywords arranged alphabetically.

Line 29. The term “hot pepper” is incorrect. The scientific name for the species Capsicum annuum is pepper. The authors should have identified the plant material used in their study using one of the taxonomic classification systems accepted worldwide. For example: https://itis.gov/servlet/SingleRpt/SingleRpt

  • The term was corrected.

Line 67. The study mentions hydric stress-related acoustic emissions. However, it is unclear what hydric stress refers to here. Even if it has been applied in another study, the method should be clearly shared.

R- Clarified the meaning of hydric stress in this context.

Line 101. In the study, hydrogen peroxide application and hydric stress-related acoustic emissions are defined as stress factors. What is the likelihood that these two different factors will affect the plant together in normal cultivation? The authors should clarify this issue.

R- Clarified the likelihood of this happening accidentally

Line 115. The photograph is not of scientific quality. Descriptions are provided from left to right. According to this, the plant on the far left is the strongest. However, no application was made to the plant on the far right, yet it is the weakest. This photograph should be reviewed.

R- Photograph have been improved in quality and we consider important to visually support the results of plant height displayed in Fig 1A..

Line 143. The statistical analysis results in Figure 4A should be reviewed. It is surprising that the control (1.5) and H2O2 (0.6) are in the same category.

R- Figure has been reviewed and corrected.

The biggest shortcoming of the study is that the applications are not clearly stated. The hydrogen peroxide (50 mM) application is understandable. However, other factors are unclear. In one part of the article, there is a mention of acoustic emission associated with moderate hydric stress. In other parts, terms such as sound stress are used. This issue should be clearly explained in both the introduction and the materials and methods section. This deficiency makes it impossible to evaluate the study as a whole.

R- Terminology corrected, the used sound stress were acoustic emissions associated with moderate hydric stress (MHAF).

In the Conclusion section, information about the limitations of the study should be added.

 The references used in the study are appropriate in terms of both subject matter and relevance.

Reviewer 3 Report

Comments and Suggestions for Authors

Model Diagram

  1. This study proposes that MHAF and H₂O₂ co-treatment triggers a series of signaling cascades leading to enhanced plant resistance. However, only a limited number of genes (e.g., Ros1, Mek1, NPR1, PR1a) were analyzed by qPCR, and the causal relationships depicted in the model have not been experimentally validated.
  2. Multiple proposed causal pathways lack experimental support. The diagram illustrates signal transduction initiated by sound and H₂O₂, ultimately leading to plant defense activation. However, only a few genes were assessed at the transcriptional level, and no protein-level evidence (e.g., p-MAPK detection) was provided. NPR1is shown as a central node in the SA pathway, yet no SA quantification or use of SA inhibitors was included to demonstrate the pathway's involvement.
  3. The diagram presents overly deterministic connections, failing to reflect data uncertainty. The use of solid arrows suggests experimentally validated pathways, but expression of PR1ashowed large variability, which is not reflected in the illustration.
  4. It is recommended to use dashed arrows or question marks to indicate hypothetical links, or to clarify in the legend that certain parts of the model are speculative. Future work should incorporate Western blotting and hormone (SA) quantification to validate key components.
  5. The current title, “Morphological, biochemical and molecular responses to hydrogen peroxide and hydric stress-related acoustic emissions in Capsicum annuumL. applied in single or combined manner,” is too broad, especially the phrase “morphological, biochemical and molecular responses,” which may mislead readers into expecting a comprehensive study covering systematic mechanisms at morphological, biochemical, and molecular levels. However, the actual research mainly focuses on expression analysis of selected genes (such as MAPK family genes and defense-related genes) and a few biochemical indicators (such as SOD and POD activities), lacking in-depth analysis of morphology and overall biochemical pathways. It is recommended to narrow the scope of the title to more accurately reflect the content of the study.

Specific Comments

  1. Line 44: The statement that peppers are mainly grown in arid regions is inaccurate. Peppers are thermophilic and drought-tolerant but not limited to arid environments.
  2. Lines 51–70: While the study introduces the use of H₂O₂ and acoustic stimulation as stressors, the rationale for their combined application in Capsicum annuum is unclear. The biological relevance and novelty of combining these two treatments are not well articulated.
  3. Figure 2: The figure lacks a scale bar, making it difficult to interpret. Control groups are not labeled, and the arrangement of pepper samples appears cluttered. It is recommended to improve figure layout and annotation for clarity.
  4. Figure 5: Terminology should be consistent throughout—avoid alternating between “sound” and “MHAF” in figure titles and labels. Please revise the entire manuscript for consistency.
  5. Figure 6: Y-axis labels should be corrected to “Phenolic,” “Flavonoid,” “DPPH,” and “ABTS.” For instance, use “Phenolic content (mg GAE/g)” for the y-axis.
  6. Figure 7: Gene names should be italicized and lowercase. Improve figure layout. Some error bars are nearly equal to or larger than the means; please indicate the number of replicates and clarify error representation.
  7. Figure 8: Gene names should be italicized and lowercase. Figure layout should also be optimized.
  8. Figure 9: In the control group, the values are too low to be visible. Please optimize the Y-axis scale for better visualization.
  9. Lines 234–237 and the corresponding results section: Discussion should begin with Figure 9(a), emphasizing the large expression changes of Ros1, before transitioning to 9(d), to enhance logical flow.
  10. Figure 10: The image quality is insufficient. Please provide a high-resolution PCA figure with labeled key variables and treatment groups.
  11. Line 264: The discussion section should be divided into thematic subsections, such as 3.1 Enzymatic antioxidants, 3.2 Non-enzymatic antioxidants, 3.3 Gene expression, and 3.4 PCA interpretation.
  12. Line 269: Only one pepper cultivar (Serrano) was tested, yet the authors make general claims. Please revise or provide justification.
  13. Line 293: The general statement "Notably, SOD, CAT and POD are important ..." should be moved before the discussion of SOD activity to improve logical flow and cohesion.

14.Lines 295–296: CAT and POD show different trends and should be discussed separately.

  1. Line 297: Figures C and D show that CAT activity decreased significantly under Sound and Sound+H₂O₂ treatments, while POD activity showed only a modest and statistically insignificant increase. Therefore, the conclusion that POD compensated for reduced CAT activity lacks sufficient support. Please revise this claim or provide further validation of antioxidant enzyme responses.

16.Line 302: A new paragraph should begin here, with a new subheading. Also, note that flavonoid content decreased, rather than increased, under combined stress conditions.

  1. Line 315: You may add the following sentence at the end of the paragraph: “Based on antioxidant indicators, the combined stress may lead to more severe oxidative damage.”
  2. Line 326: Please define “dB”; the current abbreviation is unclear.
  3. The discussion of erf1/asr1should precede that of npr1/pr1ato maintain consistency with the order in Figures 8 and 9.
  4. Lines 331–335: The description of npr1and pr1aexpression is unclear. The manuscript states that npr1 was significantly upregulated under combined treatment, while pr1a showed “no significant difference.” However, Figure 9D shows pr1a expression was clearly higher under the combined treatment than under MHAF alone and was comparable to the H₂O₂ group. This suggests a trend, or possibly a significant increase, that contradicts the statement in the text. The discussion does not address this discrepancy.
  5. Line 343: The claim “while the combined treatment had no effect compared to the control group” contradicts Figure 8, where def1expression is significantly lower under the combined treatment than in the control group. Please correct this statement.
  6. Line 346: Discussion of met1, npr1, pr1a, and rosshould follow the sequence shown in Figures 9a–d to improve structural clarity and coherence.
  7. Lines 365–375: Please provide a PCA loadings matrix or biplot to illustrate variable contributions. Include additional statistical analyses such as correlation analysis or ANOVA. PCA alone does not demonstrate synergistic effects; to validate interactions, appropriate statistical methods (e.g., interaction effect analysis) should be applied.
  8. Figure 11: Recommend moving to the Supplementary Material.
  9. Line 435: Units are inconsistent throughout the manuscript (e.g., mg vs g). Please standardize.
  10. Section 4.7: Please explain how capsaicin was detected by GC-MS without derivatization. Also, confirm whether acetonitrile was used as the injection solvent for the capsaicin standard.
  11. Table 1: Recommend placing in the Supplementary Material.
  12. Conclusion: Please include statistical analysis to support the claim of a synergistic effect (e.g., interaction effect analysis). Without such analysis, the conclusion remains unsubstantiated.
  13. Avoid overinterpreting correlation as causation (e.g., in the MAPK pathway). Only gene expression levels were measured; protein activity and phosphorylation were not assessed, and thus pathway activation cannot be confirmed. MPK4-3showed downregulation contrary to other MAPK genes, but its biological relevance (e.g., potential negative feedback regulation) was not discussed. Verification of MAPK activation through Western blotting is strongly recommended.

Author Response

Reviewer 3:

Comments and Suggestions for Authors

Model Diagram

This study proposes that MHAF and H2O2 co-treatment triggers a series of signaling cascades leading to enhanced plant resistance. However, only a limited number of genes (e.g., Ros1, Mek1, NPR1, PR1a) were analyzed by qPCR, and the causal relationships depicted in the model have not been experimentally validated.

Multiple proposed causal pathways lack experimental support. The diagram illustrates signal transduction initiated by sound and H2O2, ultimately leading to plant defense activation. However, only a few genes were assessed at the transcriptional level, and no protein-level evidence (e.g., p-MAPK detection) was provided. NPR1is shown as a central node in the SA pathway, yet no SA quantification or use of SA inhibitors was included to demonstrate the pathway's involvement.

The diagram presents overly deterministic connections, failing to reflect data uncertainty. The use of solid arrows suggests experimentally validated pathways, but expression of PR1ashowed large variability, which is not reflected in the illustration.

It is recommended to use dashed arrows or question marks to indicate hypothetical links, or to clarify in the legend that certain parts of the model are speculative. Future work should incorporate Western blotting and hormone (SA) quantification to validate key components.

The current title, “Morphological, biochemical and molecular responses to hydrogen peroxide and hydric stress-related acoustic emissions in Capsicum annuumL. applied in single or combined manner,” is too broad, especially the phrase “morphological, biochemical and molecular responses,” which may mislead readers into expecting a comprehensive study covering systematic mechanisms at morphological, biochemical, and molecular levels. However, the actual research mainly focuses on expression analysis of selected genes (such as MAPK family genes and defense-related genes) and a few biochemical indicators (such as SOD and POD activities), lacking in-depth analysis of morphology and overall biochemical pathways. It is recommended to narrow the scope of the title to more accurately reflect the content of the study.

R- Thanks for all your excellent suggestions. All these have been addressed in the revised version.

Specific Comments

Line 44: The statement that peppers are mainly grown in arid regions is inaccurate. Peppers are thermophilic and drought-tolerant but not limited to arid environments.

R- We agree. Clarified in the text.

Lines 51–70: While the study introduces the use of H2O2 and acoustic stimulation as stressors, the rationale for their combined application in Capsicum annuum is unclear. The biological relevance and novelty of combining these two treatments are not well articulated.

R- Rationale explained. These 2 stressors have been successfully tested in our research group in a single manner in several crops. This is why we consider important to evaluate the possibility of several interactions (positive or negative) when tested together. In order to obtain information that might be useful in the future possible applications of these strategies in the agricultural production of pepper.

Figure 2: The figure lacks a scale bar, making it difficult to interpret. Control groups are not labeled, and the arrangement of pepper samples appears cluttered. It is recommended to improve figure layout and annotation for clarity.

R- Figure 2 was showed only to visually support the results showed in the Fig 1A of the revised version.

Figure 5: Terminology should be consistent throughout—avoid alternating between “sound” and “MHAF” in figure titles and labels. Please revise the entire manuscript for consistency.

Figure 6: Y-axis labels should be corrected to “Phenolic,” “Flavonoid,” “DPPH,” and “ABTS.” For instance, use “Phenolic content (mg GAE/g)” for the y-axis.

Figure 7: Gene names should be italicized and lowercase. Improve figure layout. Some error bars are nearly equal to or larger than the means; please indicate the number of replicates and clarify error representation.

Figure 8: Gene names should be italicized and lowercase. Figure layout should also be optimized.

Figure 9: In the control group, the values are too low to be visible. Please optimize the Y-axis scale for better visualization.

Lines 234–237 and the corresponding results section: Discussion should begin with Figure 9(a), emphasizing the large expression changes of Ros1, before transitioning to 9(d), to enhance logical flow.

R- Thanks and agree with all your kind and excellent comments. All have been Rearranged in the revised version.

Figure 10: The image quality is insufficient. Please provide a high-resolution PCA figure with labeled key variables and treatment groups.

R- Agree again and addressed according to your suggestion in the revised version.

Line 264: The discussion section should be divided into thematic subsections, such as 3.1 Enzymatic antioxidants, 3.2 Non-enzymatic antioxidants, 3.3 Gene expression, and 3.4 PCA interpretation.

R- Subsections added.

Line 269: Only one pepper cultivar (Serrano) was tested, yet the authors make general claims. Please revise or provide justification.

R- Revised the statement.

Line 293: The general statement "Notably, SOD, CAT and POD are important ..." should be moved before the discussion of SOD activity to improve logical flow and cohesion.

R- Rearranged.

14.Lines 295–296: CAT and POD show different trends and should be discussed separately.

Line 297: Figures C and D show that CAT activity decreased significantly under Sound and Sound+H2O2 treatments, while POD activity showed only a modest and statistically insignificant increase. Therefore, the conclusion that POD compensated for reduced CAT activity lacks sufficient support. Please revise this claim or provide further validation of antioxidant enzyme responses.

R- We agree with your comment. Other explanation is the possible activity of another antioxidants not measured in the study (i.e. glutathion)

16.Line 302: A new paragraph should begin here, with a new subheading. Also, note that flavonoid content decreased, rather than increased, under combined stress conditions.

R- New paragraph added and clarified the flavonoid content.

Line 315: You may add the following sentence at the end of the paragraph: “Based on antioxidant indicators, the combined stress may lead to more severe oxidative damage.”

R- Recommended paragraph ending added.

Line 326: Please define “dB”; the current abbreviation is unclear.

R- explained the meaning of the abbreviation.

The discussion of erf1/asr1should precede that of npr1/pr1ato maintain consistency with the order in Figures 8 and 9.

R- Rearranged order.

Lines 331–335: The description of npr1and pr1aexpression is unclear. The manuscript states that npr1 was significantly upregulated under combined treatment, while pr1a showed “no significant difference.” However, Figure 9D shows pr1a expression was clearly higher under the combined treatment than under MHAF alone and was comparable to the H2O2 group. This suggests a trend, or possibly a significant increase, that contradicts the statement in the text. The discussion does not address this discrepancy.

Line 343: The claim “while the combined treatment had no effect compared to the control group” contradicts Figure 8, where def1expression is significantly lower under the combined treatment than in the control group. Please correct this statement.

Line 346: Discussion of met1, npr1, pr1a, and rosshould follow the sequence shown in Figures 9a–d to improve structural clarity and coherence.

Lines 365–375: Please provide a PCA loadings matrix or biplot to illustrate variable contributions. Include additional statistical analyses such as correlation analysis or ANOVA. PCA alone does not demonstrate synergistic effects; to validate interactions, appropriate statistical methods (e.g., interaction effect analysis) should be applied.

R- Thanks. These suggestions have been included in the PCAs figure in the revised version. ANOVA was made and inserted as supplementary material.

Figure 11: Recommend moving to the Supplementary Material.

R- Moved to supplementary material.

Line 435: Units are inconsistent throughout the manuscript (e.g., mg vs g). Please standardize.

R-Thanks. Done

Section 4.7: Please explain how capsaicin was detected by GC-MS without derivatization. Also, confirm whether acetonitrile was used as the injection solvent for the capsaicin standard.

R- The determination of capsaicin by GC-MS was assayed in various ways. First, a bibliographic search revealed that it is typically analyzed using HPLC-DAD, LC-MS, or GC-FID/GC-MS. Among reports of capsaicin determination using GC-MS, methods involving both derivatized and non-derivatized capsinoids have been described (Blaško et al., 2022; Singh et al., 2009). Therefore, we injected both the derivatized BSTFA + 1% TMCS extract and the non-derivatized extract, obtaining the results shown below. Figure 1 shows the derivatized capsaicin spectrum, and Figure 2 shows the non-derivatized capsaicin spectrum. Additionally, Figure 3 shows the chromatogram that corresponds to the derivatized extract, whereas Figure 4 shows the chromatogram of the non-derivatized extract. As can be seen, derivatization made more compounds volatile, resulting in a more complex chromatogram, but had no significant effect on capsaicin and dihydrocapsaicin resolution. Moreover, derivatization increases the cost of the analysis. For this reason, we decided to inject the extracts directly without derivatizing, and quantitate using the SIM mode for the base peak in 137.

Figure 1. Derivatized capsaicin mass spectrum from a sample injection and the first database hit.

Figure 2. Non-derivatized capsaicin mass spectrum from a sample injection and the first database hit.

Figure 3. Chromatogram from the derivatized Capsicum extract, capsaicin and dihydrocapsaicin shown in retention times 17.4 and 17.6 min, respectively.

Figure 4. Figure 3. Chromatogram from the non-derivatized Capsicum extract, capsaicin and dihydrocapsaicin shown in retention times 17.1 and 17.2 min, respectively.

Regarding the solvent used, we attempted to resuspend the extracts in both acetonitrile (figure 5) and methylene chloride (figure 6) as reported by Singh et al. in 2009, where they used GC-MS to compare against an HPLC-DAD method. However, acetonitrile produced more intense peaks for other capsinoids. Therefore, we decided to use acetonitrile for the extraction and resuspension of extracts, as well as for diluting the capsaicin and dihydrocapsaicin standards for the calibration curve. The standards for the calibration were also not derivatized to avoid altering the response factor and assure exact results.

Figure 5. Chromatogram zoom in the capsinoid zone. The extract was resuspended in acetonitrile.

Figure 5. Chromatogram zoom in the capsinoid zone. The extract was resuspended in methylene chloride.

Blaško, J., Nižnanská, Ž., Kubinec, R., Mikuláš, Ľ., Nižnanský, Ľ., Kubincová, J., Kunštek, M., Duháčková, Ľ., Hrčka, R., Kabát, J., Gabrišová, Ľ., Šidlo, J., & Szabó, A. H. (2022). Simple, fast method for the sample preparation of major capsaicinoids in ground peppers, in potato chips and chilli sauces and their analysis by GC-MS. Journal of Food Composition and Analysis, 114, 104733. https://doi.org/10.1016/j.jfca.2022.104733

‌Singh, S., Jarret, R., Russo, V., Majetich, G., Shimkus, J., Bushway, R., & Perkins, B. (2009). Determination of Capsinoids by HPLC-DAD in Capsicum Species. Journal of Agricultural and Food Chemistry, 57(9), 3452–3457. https://doi.org/10.1021/jf8040287

Table 1: Recommend placing in the Supplementary Material.

R- Moved to supplementary material

Conclusion: Please include statistical analysis to support the claim of a synergistic effect (e.g., interaction effect analysis). Without such analysis, the conclusion remains unsubstantiated.

R- Statistical analysis included.

Avoid overinterpreting correlation as causation (e.g., in the MAPK pathway). Only gene expression levels were measured; protein activity and phosphorylation were not assessed, and thus pathway activation cannot be confirmed. MPK4-3showed downregulation contrary to other MAPK genes, but its biological relevance (e.g., potential negative feedback regulation) was not discussed. Verification of MAPK activation through Western blotting is strongly recommended.

R- Clarified that evaluations such as western blot are important for future studies.

Round 2

Reviewer 1 Report

Comments and Suggestions for Authors

The author has made the required revisions, and it is recommended for acceptance.

Author Response

Thank you very much for your decision.

best

Reviewer 2 Report

Comments and Suggestions for Authors

In my assessment, it is clear that the authors have largely addressed my previous comments. As it stands, it is suitable for publication.

Author Response

Dear reviewer

thank you very much for your decision.

Best

Reviewer 3 Report

Comments and Suggestions for Authors
  • 1. Figure 1 presents statistical data on plant height, while Figure 2 shows phenotypic images of the peppers. A scale bar should be added to the photos to facilitate interpretation and to clarify the different nature of the two figures, avoiding confusion. In addition, the arrangement of the pepper plants in Figure 2 appears somewhat disordered; it is recommended to re-photograph the plants with a more organized layout and include a scale bar.

  • 2. There are two figures both labeled as Figure 2 in the manuscript. Please check and correct the numbering.

  • 3. In Figure 2, which presents the statistical analysis of capsaicin and H₂O₂, the Y-axis lacks units. It is recommended to add the appropriate units for clarity.

  • 4. In Figure 4, there is a typographical error on the Y-axis; “Conten” should be corrected to “content.”

  • 5. In Figure 6, it is suggested to rearrange the three subfigures in a single row to make the layout more uniform and facilitate comparison.

  • 6. Regarding the discussion on lines 346, it is recommended to discuss the genes ros1, met1, npr1, and pr1a in the order presented in Figures 9a–d, in order to improve structural clarity and coherence. This issue has been pointed out previously.

  • 7. Regarding the GC-MS method description, for the analysis of underivatized pure analytes, the chromatograms shown in Figures 5 and 6d appear more like full-scan (scan) data. It is recommended to provide detailed mass spectrometry parameters, specifying whether full-scan (scan) mode or selected ion monitoring (SIM) mode was used; if SIM mode was used, please indicate the monitored ions and their retention times.

Author Response

Thanks for your comments that helped to improve the writing and structure of the manuscript.

Comment 1: Figure 1 presents statistical data on plant height, while Figure 2 shows phenotypic images of the peppers. A scale bar should be added to the photos to facilitate interpretation and to clarify the different nature of the two figures, avoiding confusion. In addition, the arrangement of the pepper plants in Figure 2 appears somewhat disordered; it is recommended to re-photograph the plants with a more organized layout and include a scale bar. 

Address: Figure 2 has been rearranged so that the order of the treatments was in the same order as the ones in the graphs, a scale bar was added as well as labels of the treatments to facilitate the visualization of the figure.

Comment 2: There are two figures both labeled as Figure 2 in the manuscript. Please check and correct the numbering.

Address: The numbers of the figures was corrected accordingly.

Comment 3: In Figure 2, which presents the statistical analysis of capsaicin and H₂O₂, the Y-axis lacks units. It is recommended to add the appropriate units for clarity. 

Address: The units were added to the Y-axis of the graphs. This Figure is number 3, not number 2 in this revised 2 version.

Comment 4: In Figure 4, there is a typographical error on the Y-axis; “Conten” should be corrected to “content.”In this revised 2 version, this Figure is number 5.

Address: The typographical error was corrected.

Comment 5: In Figure 6, it is suggested to rearrange the three subfigures in a single row to make the layout more uniform and facilitate comparison.

Address: Subfigures were rearranged as suggested. This Figure is now Figure 7 in this revised version 2.

Comment 6:  Regarding the discussion on lines 346, it is recommended to discuss the genes ros1, met1, npr1, and pr1a in the order presented in Figures 9a–d, in order to improve structural clarity and coherence. This issue has been pointed out previously.

Address: The discussion of genes was rearranged in accordance to their appearance in the previous graphs. Noteworthy again to visualize that in this revised 2 version, the Figure 9 is now Figure 8.

Comment 7: Regarding the GC-MS method description, for the analysis of underivatized pure analytes, the chromatograms shown in Figures 5 and 6d appear more like full-scan (scan) data. It is recommended to provide detailed mass spectrometry parameters, specifying whether full-scan (scan) mode or selected ion monitoring (SIM) mode was used; if SIM mode was used, please indicate the monitored ions and their retention times.

Address: Your comments and suggestions are very pertinent. The acquisition mode was set to obtain both full scan and SIM spectra simultaneously using a mass range of 29–500 m/z for the SCAN mode. Nevertheless, the quantification presented in this paper was based solely on the SIM results. The SIM ion monitored to that end was the base peak m/z 137, which was significantly more intense than the rest of the ions obtained in the full scan. The SIM mode was preferred as it improves selectivity and reduces matrix interference. We decided to acquire the data in dual mode, since the width and intensity of the chromatographic peaks were sufficient for this approach. We set a 100 ms dwell time and used low resolution for the SIM mode to improve sensitivity. It is worth noting that in the last revision, we added chromatograms from the initial assays (run in 2023). Later, the resolution was optimized until we achieved the temperature gradient reported in the paper. Therefore, retention times varied, being 21.8 min for capsaicin and 22.2 min for dihydrocapsaicin, respectively, at the time we run the analyses for this research.

The materials and methods were updated according to the correct parameters discussed here, including all parameters of the mass spectrometer (L520-523), the transfer line temperature (L527), the retention time and calibration procedure (L530-532), and the rationale for not derivatizing (L 532-534).
